# Lung Protection Strategies during Cardiopulmonary Bypass Affect the Composition of Blood Electrolytes and Metabolites—A Randomized Controlled Trial

**DOI:** 10.3390/jcm7110462

**Published:** 2018-11-21

**Authors:** Katrine B. Buggeskov, Raluca G. Maltesen, Bodil S. Rasmussen, Munsoor A. Hanifa, Morten A.V. Lund, Reinhard Wimmer, Hanne B. Ravn

**Affiliations:** 1Department of Cardiothoracic Anesthesiology, Rigshospitalet, Copenhagen University Hospital, 2100 Copenhagen, Denmark; hanne.berg.ravn@regionh.dk; 2Department of Anesthesia and Intensive Care, Aalborg University Hospital, 9000 Aalborg, Denmark; bodil.steen.rasmussen@rn.dk (B.S.R.); mhanifa@dcm.aau.dk (M.A.H.); 3Department of Clinical Medicine, School of Medicine and Health, Aalborg University, 9000 Aalborg, Denmark; 4Department of Biomedical Sciences, University of Copenhagen, 2100 Copenhagen, Denmark; mortentl@sund.ku.dk; 5Department of Chemistry and Bioscience, Aalborg University, 9220 Aalborg, Denmark; rw@bio.aau.dk

**Keywords:** COPD, ischemia-reperfusion, CPB, lung protection, pulmonary perfusion, HTK, oxygenated blood, metabolites, randomized controlled trial

## Abstract

Cardiac surgery with cardiopulmonary bypass (CPB) causes an acute lung ischemia-reperfusion injury, which can develop to pulmonary dysfunction postoperatively. This sub-study of the Pulmonary Protection Trial aimed to elucidate changes in arterial blood gas analyses, inflammatory protein interleukin-6, and metabolites of 90 chronic obstructive pulmonary disease patients following two lung protective regimens of pulmonary artery perfusion with either hypothermic histidine-tryptophan-ketoglutarate (HTK) solution or normothermic oxygenated blood during CPB, compared to the standard CPB with no pulmonary perfusion. Blood was collected at six time points before, during, and up to 20 h post-CPB. Blood gas analysis, enzyme-linked immunosorbent assay, and nuclear magnetic resonance spectroscopy were used, and multivariate and univariate statistical analyses were performed. All patients had decreased gas exchange, augmented inflammation, and metabolite alteration during and after CPB. While no difference was observed between patients receiving oxygenated blood and standard CPB, patients receiving HTK solution had an excess of metabolites involved in energy production and detoxification of reactive oxygen species. Also, patients receiving HTK suffered a transient isotonic hyponatremia that resolved within 20 h post-CPB. Additional studies are needed to further elucidate how to diminish lung ischemia-reperfusion injury during CPB, and thereby, reduce the risk of developing severe postoperative pulmonary dysfunction.

## 1. Introduction

In cardiac surgery, the use of cardiopulmonary bypass (CPB) deprives the pulmonary arteries of blood flow, with the lung tissue reliant on blood supply from the bronchial arteries. Therefore, the lungs are at risk of relative ischemia during CPB, followed by a reperfusion injury after weaning from CPB, which may cause impaired postoperative pulmonary function [1,2], particularly in patients with preexisting lung disease. In this respect, patients with chronic obstructive pulmonary disease (COPD) are more susceptible to CPB associated lung injury [3,4,5], as they are more vulnerable to the ceasing of perfusion and ventilation [6,7,8]. The reduced pulmonary perfusion during CPB is accompanied by a series of metabolic changes and bypass duration impacts not only the metabolic activity [9], but also the fibrinolytic cascade [10]. Eventually, these events lead to pulmonary dysfunction with subsequent hypoxemia, and in the most severe cases to development of acute respiratory distress syndrome [11,12,13]. Studies of acute respiratory failure have identified a characteristic hallmark of lung injury with overall energy depletion [14], activated lipid metabolism, augmented anaerobic environment, proteolysis, oxidative stress [15], apoptosis [16], and endothelial disruption [17].

Previous studies suggest that pulmonary artery perfusion during CPB may diminish ischemia-reperfusion injury, and subsequent pulmonary dysfunction, by supplying metabolic substrates to the lungs [18,19,20,21] and by reducing cellular oxygen consumption [22,23,24,25,26,27,28]. A wide variety of protective solutions are routinely used, however, there is an ongoing discussion about the relative effectiveness of these solutions with regards to tissue protection during an ischemia-reperfusion period [29]. Warm or cold autologous blood [22,23,24,25,26,27,28] and crystalloids [18,19,20,21] are the main preservative solutions used in cardiac surgery. The crystalloid solutions can be classified into those based on extracellular components with high potassium, magnesium, and bicarbonate levels; and those based on intracellular electrolytes. The hypothermic histidine-tryptophan-ketoglutarate (HTK) solution is an extracellular based preservation solution used for organ (e.g., lung) transplantation [21,30,31,32], in addition to its use as a cardioplegic agent. Preservation with HTK-Custodiol (Koehler Chemi, GMBH, Germany), or its equivalent Bretschneider solution [33,34,35], is administered as a single dose and claimed to give organ protection for more than two hours [36]. The solution contains 15 mM Na^+^, 9 mM K^+^, 4 mM Mg^++^, 0.015 mM Ca^++^, 198 mM histidine, 2 mM tryptophan, 1 mM 2-ketoglutarate, and 30mM mannitol, with a pH of 7.02–7.2 at 25 °C (77 °F) [37]. The included histidine is thought to have a buffer effect associated with enhanced efficiency of anaerobic glycolysis; tryptophan and mannitol are proposed to function as cell membrane stabilizers and as reducing agents to prevent the formation of cellular edema; whereas 2-ketoglutarate is an intermediate of the tricarboxylic acid (TCA) cycle and a precursor of nicotinamide adenine dinucleotide which is assumed to serve in energy production [36]. Several studies have demonstrated the efficacy of HTK solution in preserving myocardial adenosine triphosphate stores, improving post-arrest contractile function and minimizing myocardial necrosis [38,39]. Careaga and colleagues [40] reported that patients receiving cardioplegia with HTK solution had a decreased incidence of postoperative arrhythmias, length-of-stay in the intensive care unit, and inotropic support; while Lin and colleagues [41] have shown that blood transfusions, mortality, morbidity, and hospital readmission were significantly lower compared to those receiving another extracellular based cardiologic solution (St. Thomas’ Hospital solution, 144 mEq Na^+^, 20 mEq K^+^, 32 mEq Mg^++^, and 4.8 mEq Ca^++^ per liter solution, pH 5.5).

In an experimental rat model of ex vivo lung perfusion, it was found that HTK solution attenuated pulmonary edema until 12 h post-perfusion compared to both saline and low-potassium dextran solution, another extracellular based preservation solution [42,43]. We have recently shown that lung ischemia-reperfusion injury during CPB is characterized by augmented inflammation, metabolic acidosis, protease activity, and oxidative stress and that patients receiving lung protection with HTK solution seemed better protected against severe acidosis, excessive fatty acid oxidation, and inflammation compared to pulmonary artery perfusion with normothermic oxygenated blood or no pulmonary perfusion during CPB [44]. Besides HTK, pulmonary artery perfusion with normothermic oxygenated blood during CPB has also been found to reduce lung ischemia-reperfusion injury with improved postoperative oxygenation and lung compliance, and reduced intubation time [22,26,27,45]. In the original paper, from the randomized Pulmonary Protection Trial of adult patients undergoing elective cardiac surgery [46], we found that postoperative arterial oxygenation was higher 20 h post-CPB in patients receiving pulmonary perfusion with oxygenated blood during CPB, compared to those receiving HTK solution or standard CPB without pulmonary perfusion. However, baseline mean arterial oxygenation was 20% higher in the patients receiving pulmonary perfusion with oxygenated blood compared to the two other groups baseline values [46].

Currently, no randomized studies have compared the effects of pulmonary artery perfusion with HTK solution or oxygenated blood, versus standard CPB, and little is known about the molecular mechanisms of these regimens. Hence, the main aim of this sub-study is to compare the molecular effects of hypothermic HTK solution and normothermic oxygenated blood to the standard CPB regimen without pulmonary perfusion, on blood obtained from COPD patients included in the randomized Pulmonary Protection Trial [46]. In particularly, we aim to find changes in arterial blood gases, electrolytes, plasma inflammatory protein interleukin-6 (IL-6), and plasma metabolites associated with ischemia-reperfusion injury and the received regime. Blood samples were collected from 90 patients undergoing cardiac surgery at baseline (pre-CPB), immediately after weaning from CPB (end CPB), and the following 2, 4, 6, and 20 h after weaning from CPB. Profound blood gas, electrolyte, inflammatory protein, and metabolite changes were detected during and after CPB in all patients, regardless of the treatment received. While all patients had decreased gas exchange, augmented inflammation, and metabolite alterations, patients receiving HTK solution had an excess of metabolites involved in energy production and detoxification of reactive oxygen species. In addition, patients receiving HTK suffered a transient isotonic hyponatremia that resolved within 20 h post-CPB.

## 2. Materials and Methods

The Pulmonary Protection Trial (ClinicalTriAls.gov Identifier: NCT01614951) was conducted in accordance with the Declaration of Helsinki and approved by the Biomedical Research Ethics Committees of The Capital Region of Denmark (H-1-2012-024), the Danish Medicines Agency (2012024017; EudraCT, 2011-006290-25, 4141), and the Danish Data Protection Agency (2011-41-7051). The rationale for, design, and primary results of the trial have been previously published [44,46,47].

### 2.1. Patient Population, Interventions, and Sample Collection

After obtaining informed consent, 90 patients scheduled for elective coronary artery bypass grafting, aortic valve replacement, or both of the aforementioned procedures were included. Inclusion criteria were adults (≥18 years) with COPD. Exclusion criteria were patients with previous heart or lung surgery, previous thoracic radiation, left ventricular ejection fraction <20%, tracheal intubation, and/or medically treated pneumonia prior to surgery. Patients underwent an overnight fast, standardized anesthesia and surgical management, and were assigned to receive the standard CPB regimen or pulmonary artery perfusion, with either continuous normothermic oxygenated blood during CPB or a single-shot hypothermic of HTK solution at the same time as the first cardioplegia (Figure 1).

Immediately after anesthetic induction, a catheter was inserted in the pulmonary artery and blood samples for interleukin-6 (IL-6) and metabolite analysis were drawn at baseline (pre-CPB), immediately after weaning from CPB (end CPB), and the following 2, 4, 6, and 20 h after weaning from CPB. Samples were simultaneously drawn from a catheter in arteria radialis for immediate blood gas analysis. The 768 blood samples from the pulmonary artery were immediately centrifuged at 4 °C to obtain plasma. Aliquots were stored at −80 °C until analysis.

### 2.2. Interleukin-6

Plasma IL-6 levels were determined using a commercial enzyme-linked immunosorbent assay (ELISA) kit (Human IL-6 ELISA Ready-SET-Go; eBioscience, San Diego, CA, USA) according to the manufacturer’s protocol. Samples were run in duplicate, and the mean was obtained.

### 2.3. Nuclear Magnetic Resonance (NMR) Spctroscopy

Plasma samples were thawed for 30 min at 4 °C and centrifuged for 5 min (4 °C, 12,100× g) to remove cells and precipitate. Aliquots of 400 μL of supernatant were mixed with 200 µL D_2_O phosphate buffer (0.2 M Na_2_HPO_4_/NaH_2_PO_4_, pH 7.4, in 99% ^2^H_2_O; Sigma-Aldrich, Germany) and subsequently recorded on a BRUKER AVIII-600 MHz NMR spectrometer (BrukerBioSpin, Germany) equipped with a CPP-TCI probe (Bruker BioSpin, Switzerland) at a temperature of 298.1 K. ^1^H NMR spectra were acquired using a T_2_-filtered one-dimensional Carr–Purcell–Meiboom–Gill (CPMG) [48] pulse sequence with water suppression, as previously described [9,15]. Spectral acquisition was controlled using the TopSpin 3.1 software (Bruker BioSpin, Germany).

### 2.4. Data Analysis

Spectra were manually phased and baseline corrected, calibrated to the methyl peaks of L-alanine at 1.48 ppm, and regions containing water and ethylenediaminetetraacetate EDTA signals were removed. Data was reduced to equal buckets of 0.001 ppm bin widths, generalized log transformed, normalized, and mean centered. Analysis was performed in the matrix laboratory MATLAB (R2011b, MathWorks, Natick, MA, USA) and IBM® SPSS Statistics (v.22, SPSS Inc., Armonk, NY, USA). For multivariate analysis unsupervised principal component analysis was applied using the Partial Least-Square PLS-Toolbox 6.5 (Eigenvector Research, Wenatchee, WA, USA). Signals contributing to sample clustering were identified and quantified as previously described [9,15]. Concentrations are presented as mean ± standard deviation and percent change, calculated by the formula (postCPB−preCPB)/preCPB∙100.

The interactions between time- and treatment-dependent changes were determined by two-way analysis of variance (ANOVA) with Tukey’s post-hoc test for multiple comparisons. Statistical significance was defined as a *p*-value ≤0.05.

## 3. Results

### 3.1. Patient Characteristics

A total of 90 patients were randomized to receive either the standard CPB regimen with no pulmonary perfusion, continues pulmonary artery perfusion with normothermic oxygenated blood, or single-shot pulmonary perfusion with HTK solution. There was one post-randomization exclusion where cross-clamping to initiate CPB was not possible due to a fragile aorta. For this sub-study, the per-protocol population consisted of 34 patients receiving standard CPB, 28 receiving pulmonary perfusion with oxygenated blood, and 27 receiving pulmonary perfusion with HTK solution. Patient characteristics, medical history, preoperative pulmonary function, and surgical data are presented in Table 1. Eighty-seven patients were Caucasians, one Inuit, and one of Arabic origin. There were no statistically significant differences between the groups.

### 3.2. Clincal Outcomes, Arterial Blood Gas Analyses, and Inflammatory Protein

All clinical outcomes have previously been published [46] and later included in a systematic review [49]. No significant differences were found between the three groups with respect to postoperative oxygenation, intubation time, days alive outside the intensive care or hospital, 90-day mortality, and serious adverse events.

Arterial blood gas analyses are presented in Table 2. At the end of CPB, pH levels dropped slightly in all patients, with more pronounced decreases in the HTK-receiving patients, due to a relatively lower pH of the received solution (7.02–7.20). Base excess and bicarbonate also decreased with unchanged partial pressure of carbon dioxide in accordance with the HTK driven metabolic acidosis although without changes in lactate. At 20 h post-CPB, pH levels returned to pre-surgical values in all patients. The partial pressure of carbon dioxide was in all patients within normal range during the whole observation period. The partial pressure of oxygen was, in all patients, above physiological values from pre-CPB until normalization 20 h post-CPB. Likewise, arterial blood glucose and lactate levels increased right after CPB in all patients regardless of the treatment received, and remained elevated through the following 20 h post-CPB. Lastly, sodium and potassium also changed post-CPB; while potassium increased, sodium decreased immediately after CPB, especially in the HTK-receiving patients, and nearly returned to their baseline levels 20 h after CPB.

The systemic inflammatory response, generated by contact activation of cells passing through the extracorporeal CPB circuit in addition to the ischemia-reperfusion injury, was assessed by the pro-inflammatory mediator IL-6 (Table 2). A steep increase was observed in all patients immediately after weaning from CPB and its concentration remained elevated through all post-CPB measurements.

### 3.3. Plasma Metabolites

Figure 2 illustrates the aliphatic region of one-dimensional ^1^H-NMR spectra of median plasma samples collected before CPB (blue), at the end of CPB (cyan), 2 h post-CPB (green), 4 h post-CPB (brown), 6 h post-CPB (orange), and 20 h post-CPB (yellow). Visual inspection of spectra reveals that the intensity of several metabolites changed significantly in the peri- and postoperative period, reflecting alterations in their levels as a consequence of surgery.

Principal component analysis revealed sample clustering according to sample time (principal component one, PC1 = 36.9%, Figure 3A) and treatment received during CPB (PC2 = 17.6%, Figure 3B). As observed in Figure 3A, samples collected before (blue) and at the end of CPB (cyan) clustered separately from each other, indicating that the ischemia-reperfusion injury itself had a significant impact on the metabolome. Longer duration of CPB resulted in greater increases of anaerobic (i.e., lactate; Pearson correlation coefficient *r* = 0.4, *p* < 0.0001) and TCA cycle (i.e., citrate; *r* = 0.4, *p* < 0.0001) metabolites, and greater dilution of plasma fatty acids and lipoproteins (*r* = −0.32, *p* < 0.0001) (Figure 3C). Samples collected 2–6 h post-CPB tended to return towards baseline, and at 20 h post-CPB samples almost reached their corresponding pre-CPB values (Figure 3A). When coloring principal component analysis scores by the intervention received, HTK patients clustered separately on the PC2 axis at the end of CPB, while patients receiving normothermic oxygenated blood overlapped with patients receiving standard CPB, indicating similar metabolite changes in the oxygenated blood and standard CPB groups. A total of 41 identified metabolites were found to contribute to sample clustering along PC1 and PC2 (Appendix A. Overview of metabolite changes with time in each corresponding group), of which several were found to be influenced by both the surgery itself and the received treatment (Figure 3D).

An overview of all metabolites is provided in Figure 3E and their pathways in Figure 4. Several metabolites involved in glycolysis, amino acid metabolism, ketone synthesis, urea cycle, TCA cycle, protein and fatty acid metabolism, and glucose-derivate metabolism were found to be affected after CPB. Among these, glycolytic metabolites (e.g., glucose, myo-inositol, fucose, mannose, phosphoenoylpyruvate, pyruvate, and lactate), urea cycle metabolites (e.g., arginine, citrulline, proline, and urea), several amino acids (e.g., lysine, valine, leucine, tyrosine, phenylalanine, and creatinine), and ketone bodies (e.g., acetoacetate, acetone, and 3-hydroxybutyric acid) were all increased immediately after CPB in all patients, irrespective of the received intervention (Figure 4). The protein derived N-acetyl-glycoprotein fragments, lipoproteins, and fatty acids (e.g., free fatty acids, monounsaturated fatty acids, and triacylglycerol) were decreased in all patients at the end of CPB, due to hemodilution by the CPB-pump priming solution, with the lowest values for the HTK-receiving patients, who received an additional volume with of HTK solution. HTK-receiving patients had higher mannitol levels at the end of CPB compared to the standard and oxygenated blood receiving patients, due to the additional mannitol from the HTK solution. Increased levels of the drug clearance metabolite glucuronic acid and its precursor, myo-inositol, were observed in all patients from 2 to 20 h post-CPB, with marginally higher levels in the HTK group. After administration of HTK, plasma histidine levels increased markedly, from a physiological value of 0.068 mM to 6.4 mM at the end of CPB and were still 7-times above physiological values 20 h post-CPB (0.47 mM). In comparison, histidine levels in the standard regimen and oxygenated blood groups did not change more than 10% during the same period. Within the same interval, plasma glutamate, glutamine, aspartate, threonine, methionine, creatine, glycine, alanine, format, and 2-ketoglutarate were all elevated in the HTK group compared to the standard and oxygenated blood groups. Lastly, citrate levels increased to a minor degree postoperatively in the HTK-receiving patients compared to the patients receiving oxygenated blood or standard CPB.

## 4. Discussion

To the best of our knowledge, this is the first study to investigate the metabolic response in COPD patients undergoing cardiac surgery randomized to receive either pulmonary artery perfusion with oxygenated blood or HTK solution, compared to standard CPB regimen without pulmonary perfusion. The main aim of this sub-study was to investigate the molecular effects of the two pulmonary perfusion regimes compared to standard CPB, on blood obtained from COPD patients included in the randomized Pulmonary Protection Trial [46]. In particularly we aimed to find changes in arterial blood gas, electrolytes, plasma inflammatory protein IL-6, and plasma metabolites associated with hypothesized ischemia-reperfusion injury and the received regimen. The results should be interpreted with respect to some important limitations. Although cardiac surgical patients are relative homogenous, outcomes are, besides the interventions, influenced by the individual patient’s comorbidities, which may leave the study underpowered to detect any group differences. In addition, our follow-up period was limited to 20-h post CPB, which is not the time of maximal pulmonary damage, which questions the generalizability of our results.

### 4.1. Clincal Outcomes, Arterial Blood Gas Analyses, and Inflammatory Protein

As previously published, we found no significant differences in any of the clinical outcomes between the three groups [46,49]. Overall, arterial blood gas analyses were affected by surgery in all three groups. We found that sodium changed more significantly in the HTK receiving patients at the end of CPB. This is in line with previous findings by Lindner et al. [50] describing the development of isotonic hyponatremia following cardioplegia with HTK solution. They concluded that hyponatremia should not be corrected without concomitant hypotonicity, as it may, in fact, cause a significant increase in serum osmolarity, which can be associated with neurological complications [41]. Hyponatremia in the presence of unchanged osmolarity was also found by Viana and colleagues [37], following cardioplegia with HTK solution, which supports our findings and the strategy of not correcting the transitory electrolyte changes following administration of HTK solution.

Cardiac surgery with CPB is known to trigger a significant postoperative systemic inflammatory response, which may result in multiple-organ dysfunction associated with a poor clinical outcome [51]. The proinflammatory mediator IL-6 has previously been found to be increased after CPB, reaching peak levels 4–6 h after weaning from CPB and to decrease to normal levels within 24–48 h [52,53,54]. Our results confirm this increase in IL-6 levels following CPB, with peak levels reached at 2–6 h post-CPB. Although the HTK receiving patients in general had lower plasma IL-6 levels compared to the oxygenated blood and standard CPB regimen groups, there was no statistically significant difference between groups. These results are in line with our previous finding in bronchoalveolar fluid [44], where we found an increased production of inflammatory cells in the lungs of these patients, irrespective of the treatment received. Interestingly, at our last measurement 20 h post-CPB IL-6 levels had not declined as expected [52], presenting levels as high as those measured two hours post-CPB. Patients with COPD have previously been reported to exhibit a dysfunctional immune response compared with healthy control subjects [55,56]. How the immune response following the insult of cardiac surgery and CPB differs in patients with COPD is largely unknown, but our results suggest an altered time course with a slower resolution of inflammation.

### 4.2. Plasma Metabolites

Several metabolites changed significantly in the peri- and postoperative periods, with the largest differences between the pre- and post-CPB samples. The patients’ metabolomes tended to return towards preoperative state at the last measurement (20 h post-CPB); however, several metabolite levels were still altered at this time point. Collecting samples from later points could have demonstrated the exact time when the metabolome regained its normal preoperative level. Several metabolites correlated with the duration of CPB, indicating that a prolonged surgical procedure influenced the metabolite levels, with increased anaerobic metabolism and decreased levels of fatty acids, lipids, and glycoproteins. This is in line with previously published results where we also found a correlation between the same metabolites and the duration of CPB in patients undergoing elective coronary artery bypass grafting [9].

Major surgery leads to a general surgical stress response nearly doubling patients’ energy requirements. Easily accessible carbohydrates and fats are used first; while muscle catabolism yielding branched chain amino acids are used later on to serve as carbon chains for the unmet energy needs, with valine being converted to carbohydrates, leucine to fats, and isoleucine to fats as well as carbohydrates [57,58,59,60]. In our study the levels of free fatty acids, lipoproteins, monounsaturated fatty acids, triacylglycerol, and N-acetyl-glycoprotein fragments were significantly lower at the end of CPB. Their levels increased in the hours following surgery, but without reaching baseline levels. The drop in the N-acetyl-glycoprotein and lipoprotein macromolecules was more pronounced in the HTK receiving patients, compared to those receiving pulmonary perfusion with oxygenated blood or standard CPB. This can be explained by the possible entrapment of some of the macromolecules in the cell saver used to collect the administered HTK solution. Analyzing samples from the cell saver would confirm or reject this statement. The levels of carbohydrates and branched chain amino acids increased in all patients at the end of CPB and their levels almost normalized within 6–20 h post-CPB, with some exceptions.

Overall, we found no significant metabolite differences in patients receiving pulmonary artery perfusion with oxygenated blood compared to those receiving the standard regimen. Not even lactate, which, theoretically, should be reduced in patients receiving additional perfusion of the pulmonary circulation with oxygenated blood. This is in line with our recently published results involving metabolites measured in bronchoalveolar fluid before and after CPB, where we also found no significant differences in the lactate levels among patients [44].

Previous work has shown that reduced oxygen concentrations during ischemia enhance polyamine metabolism [61]. When compared with preoperative concentrations, the polyamine metabolite arginine and its precursors—citrulline and proline—were increased in all three groups, indicating increased oxidative stress. In line with these findings, we have recently demonstrated an excess production of polyamine metabolites in the lungs of these patients at the end of CPB (>40-fold) [44], indicating that reduced oxygenation during CPB not only affects the lungs, but also has a systemic effect.

The TCA cycle intermediate 2-ketoglutarate is known to be consumed rapidly during ischemia, since it both serves in ATP production [36] and detoxification of the increased amount of reactive oxygen species (ROS) produced [62]. We observed a steep drop in its levels immediately after weaning from CPB in the oxygenated blood and standard regimen groups, while its levels increased in the HTK receiving patients. This increase can be explained by the addition of 2-ketoglutarate through the HTK solution, and by a possible diffusion across the lungs into the systemic circuit. In the bronchoalveolar fluid we found no change in the levels of 2-ketoglutarate and its precursor, succinate, at the end of CPB in the HTK receiving patients, while their levels increased significantly in the oxygenated blood and standard regimen patients [44]. This indicates that the body may have a mechanism to mobilize the necessary substrates from the system to the organs experiencing severe energy depletion, and vice versa. In the hours following CPB, the levels of 2-ketoglutarate dropped in the HTK group as well; however, it never reached levels similar to the oxygenated blood and standard regimen groups. The TCA cycle metabolite citrate was found to be elevated immediately after CPB in all patients, with lower increases in the HTK group, indicating that 2-ketoglutarate provided some of the required energy and protection against ROS formation. It is well known that ischemia-reperfusion injury and the contact of blood with the extracorporeal CPB circuit activates the systemic inflammatory response, which in turn generate excessive ROS [52]. Hence, it can be speculated that an increased concentration of 2-ketoglutarate in the HTK solution could further minimize the production of citrate during CPB, which in turn would ameliorate the production of ROS.

Changes in metabolites involved in nitrogen elimination via nitrogen metabolism and the urea cycle were characteristic for patients receiving pulmonary artery perfusion with HTK solution during CPB. Histidine and the metabolites involved in histidine catabolism (creatine, alanine, glycine, glutamine, glutamate, threonine, and aspartate) were found to be significantly higher up to and including the last measurement, at 20 h post-CPB. Histidine rose from its physiological pre-CPB value of 0.068 mM to 6.4 mM at the end of CPB, due to HTK solution spillover from the pulmonary to the systemic circuit, and reperfusion of the pulmonary circuit after cross-clamp release during reestablishment of the patients own circulation. The byproduct of histidine and histidine-related metabolites is urea. Plasma urea increased in all patients post-CPB and the HTK receiving patients experienced elevated production four hours post-CPB. These findings are in line with previous studies performed on plasma and urine from patients receiving cardioplegia with Bretschneider solution [42,43], which demonstrated high levels of urea and histidine in both plasma and urine until 72 h post-CPB. Previous work conducted on histidine administration to cardiac surgical patients indicates a potential benefit of the intraoperative provision of the amino acids arising from its multiple conversion pathways. For example, histidine administration mitigates the energy consuming systemic inflammatory response, which develops postoperatively in all patients undergoing cardiac surgery with CPB; and it also spares endogenous protein sources, like skeletal muscle, from degradation [42,43]. The potential benefit of amino acid administration is also described by Umenai et al. [63] who showed positive effects of perioperative amino acid infusion in patients undergoing off-pump CABG, resulting in significantly shorter durations of postoperative mechanical ventilation and time in the intensive care unit. Lastly, we have previously demonstrated that histidine administration during CPB may have a lung protective effect against severe acidosis, excessive fatty acid oxidation, and inflammation [44]. These results were performed on bronchoalveolar fluid and the next step will be to investigate how HTK solution affects the composition of lung tissue.

### 4.3. Strengths and Limitations

Metabonomics represents an advance over other system-wide approaches, yet several limitations are relevant to the interpretation of our study. Firstly, hypoxemia and acute respiratory distress syndrome are common postoperative pulmonary dysfunctions following cardiac surgery, which develop 48–72 h after the end of surgery, with gradual improvement within several days [2,64]. Due to the limited follow-up of our study (20 h post-CPB), we could not assess whether the use of pulmonary artery perfusion regimens could have prevented the development of these postoperative complications. Clinical and metabolite outcomes were not measured at the time of maximal damage, which impairs the generalizability of our observations. These observations should therefore be interpreted as the first steps towards a more detailed description of pulmonary injury on the second to third postoperative days. Secondly, conducting clinical trials with a cohort of complex critically ill patients, where outcomes will be determined primarily by the severity of their underlying illness, is challenging. Although cardiac surgical patients are a relatively homogeneous group, the inevitable heterogeneity may leave this sub-study underpowered to detect group differences in relation to clinical outcomes and metabolite estimates, and specifically, for the comparison of pulmonary artery perfusion with oxygenated blood to standard CPB.

## 5. Conclusions

In this descriptive sub-study of the Pulmonary Protection Trial we found that arterial blood gas analyses, IL-6, and several metabolites changed significantly in the peri- and postoperative period in cardiac surgery patients with COPD. When comparing the molecular and electrolyte differences between pulmonary artery perfusion with oxygenated blood to the standard regimen, similar patterns were observed during and after CPB. These observations indicate that the additional blood supply does not, as hypothesized, alter what is already achieved by the bronchial arteries. When comparing the differences between pulmonary artery perfusion with HTK solution to oxygenated blood or the standard regimen, we found that some of the HTK solution enters the systemic circuit and elicits a significant impact on arterial blood gas analyses and metabolites, with some of the effects still present 20 h after its administration. Results confirm the development of an isotonic hyponatremia following administration of HTK solution, and that this should not be corrected without concomitant hypotonicity. In addition, although plasma IL-6 levels increased in all patients during and after CPB, HTK receiving patients had, in general, lower levels of this inflammatory protein; however, the difference was not statistically significant. Metabolite data indicate that pulmonary perfusion with HTK during CPB may provide some of the necessary substrates needed to counteract ischemia-reperfusion injury, protect against severe energy depletion, and ameliorate ROS production. Additional studies are needed to confirm the metabolite results and further elucidate how to diminish ischemia-reperfusion injury of the lungs during CPB to reduce the risk of developing acute respiratory distress syndrome.

## Figures and Tables

**Figure 1 jcm-07-00462-f001:**
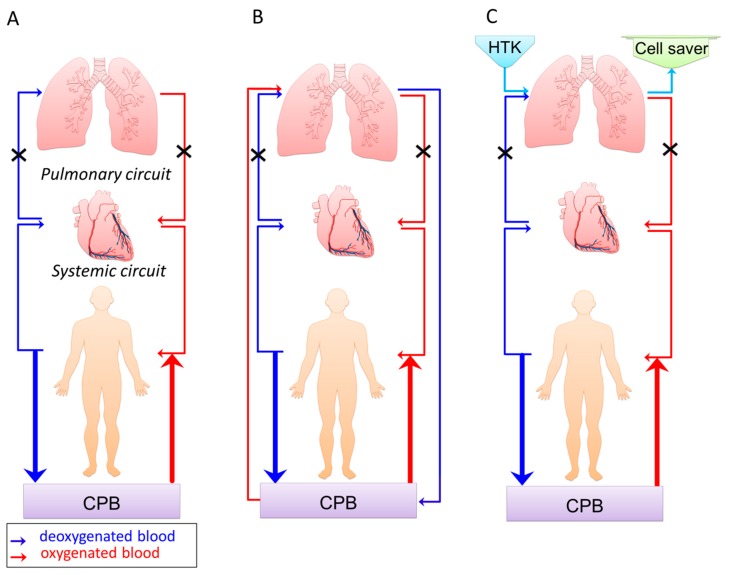
Description of the three interventions: (**a**) standard cardiopulmonary bypass (CPB) without perfusion of the pulmonary circuit; (**b**) CPB plus continuous pulmonary artery perfusion with normothermic oxygenated blood (400 mL/min); (**c**) CPB plus pulmonary artery perfusion with a single-shot of histidine-tryptophan-ketoglutarate (HTK) solution (2000 mL). The volume returning to the left atrium from the pulmonary circuit was filtered by a cell saver, only returning the red blood cells to the heart and lung machine’s venous reservoir. At the end of CPB, reperfusion of the pulmonary circuit added approximately 500 mL HTK solution to the systemic circuit.

**Figure 2 jcm-07-00462-f002:**
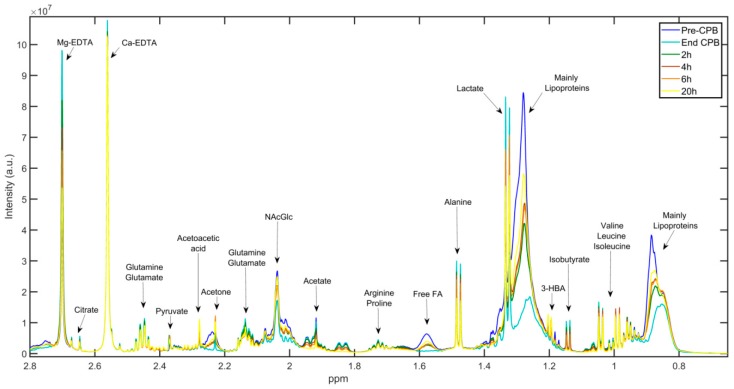
600 MHz ^1^H-NMR CPMG spectra of metabolite signals identified in plasma obtained from the standard group before and after CPB (pre-CPB, blue; end CPB, cyan; 2 h post-CPB, green; 4 h post-CPB, brown; 6 h post-CPB, orange; 20 h post-CPB, yellow). EDTA, Ethylenediaminetetraacetic acid; FA, fatty acid; 3-HBA, 3-hydroxybutyric acid; ppm, part per million; a.u., arbitrary units.

**Figure 3 jcm-07-00462-f003:**
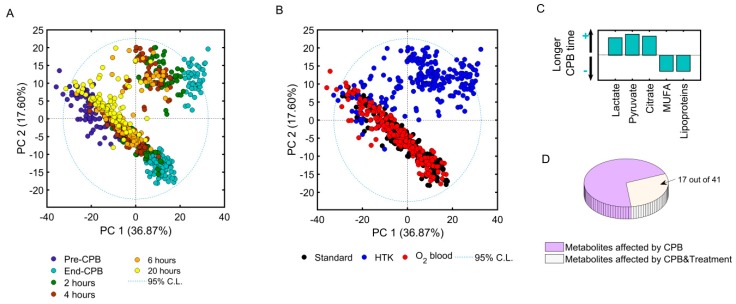
Metabolome changes as a consequence of the surgical procedure and lung protection regimen received. (**A**) Principal component analysis (PCA) scores plot containing NMR spectral data of plasma collected before CPB (blue), at the end of CPB (cyan), 2 h (green), 4 h (brown), 6 h (orange), and 20 h post-CPB (yellow). Samples cluster according to the time of sample collection. (**B**) Same PCA scores plot as in (A) but samples are color coded according to intervention group: standard CPB with no pulmonary perfusion (black), pulmonary artery perfusion with hypothermic histidine-tryptophan-ketoglutarate (HTK) (blue), pulmonary artery perfusion with normothermic oxygenated blood (O_2_ blood) (red). CL, confidence level; PC, principal component 1 and 2. (**C**) Correlation of metabolites concentrations measured at the end of CPB with CPB duration; lactate, pyruvate, and citrate levels increased (+), while monounsaturated fatty acids (MUFA) and lipoprotein levels decreased (−, with a longer duration of CPB. (**D**) Pie plot representation of metabolites demonstrated significant changes over time as a consequence of CPB (purple) and treatment allocation (white) explored by means of two-way ANOVA. (**E**) Heat map representation of mean percent changes of metabolites in relation to pre-operative values. Mean percent changes were calculated using the formula (postCPB − preCPB)/preCPB·100. Green: lower concentration; red: higher concentration. 3-HBA, 3-hydroxybutyric acid; NAc-Glc, N-acetylated glycoprotein fragment; TAG, triacylglycerol; FA, fatty acid; MUFA, monounsaturated fatty acid.

**Figure 4 jcm-07-00462-f004:**
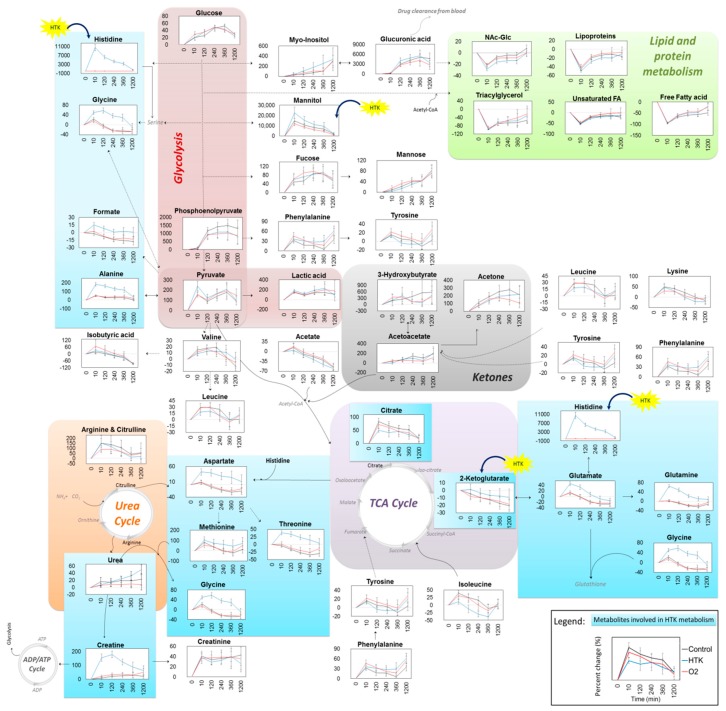
Overview of plasma metabolite changes as a consequence of the surgical procedure and treatment received (black, standard CPB; red, oxygenated blood; blue, HTK solution). To avoid baseline effects due to patient heterogeneity, each patient is considered to be their own control and data is presented as percent change from pre-CPB values in each individual patient; i.e., (postCPB − preCPB)/preCPB·100. Results are presented as means and 95% confidence intervals of the percent change. Metabolites directly affected by the HTK solution are highlighted in blue boxes. HTK solution contains histidine, tryptophan, 2-ketoglutarate, and mannitol, and these metabolites are marked with yellow HTK symbols, with the exception of tryptophan, which was not detected on the NMR spectra.

**Table 1 jcm-07-00462-t001:** Clinical and procedural characteristics (per-protocol population).

	Standard CPB (*n* = 34)	O_2_ Blood (*n* = 28)	HTK Solution (*n* = 27)
Patient characteristics			
Age—year (mean ± SD)	70.2 ± 8.5	69.6 ± 10.7	69.8 ± 8.3
Male sex—number (%)	28 (82.4)	21 (75.0)	18 (66.7)
BMI (mean ± SD)	22.9 ± 4.2	23.1 ± 5.2	22.8 ± 4.0
Medical history—number/total number (%)			
Self-reported COPD	21/34 (61.8)	14/28 (50.0)	17/27 (63.0)
Pulmonary hypertension ^†^	7/26 (27.0)	3/23 (13.0)	6/24 (25.0)
Arterial hypertension	24/34 (70.6)	22/28 (78.6)	23/27 (85.2)
Chronic atrial fibrillation or flutter	5/34 (14.7)	4/28 (14.3)	3/26 (11.5)
Chronic heart failure ^‡^	10/32 (31.3)	12/28 (42.9)	6/27 (22.2)
ASA class IV or worse	6/32 (18.8)	1/28 (3.6)	2/25 (8.0)
Left ventricular ejection fraction (IQR)	50.0 (45.0–60.0)	55.0 (45.0–60.0)	50.0 (40.0–60.0)
Recent AMI ^§^	10/34 (29.4)	6/28 (21.4)	9/27 (33.3)
Diabetes mellitus	9/34 (26.5)	8/28 (28.6)	3/27 (11.1)
Estimated creatinine clearance (IQR) **	86.6 (70.0–106.0)	81.4 (61.8–122.9)	82.7 (62.7–112.8)
Previous transient ischemic attack or stroke	7/34 (20.6)	1/28 (3.6)	7/27 (26.0)
Previous percutaneous coronary intervention	6/34 (17.7)	4/28 (14.3)	5/27 (18.6)
Implantable cardioverter defibrillator or pacemaker	1/32 (3.1)	2/27 (7.4)	1/26 (3.9)
Excess alcohol consumption ^††^	7/34 (20.6)	11/28 (39.3)	7/27 (26.0)
Tobacco pack years (IQR)	35.0 (22.5–47.5)	35.0 (17.8–50.0)	40.0 (30.0–50.0)
Pulmonary function on admission			
Percent predicted FEV1 (IQR)	73.5 (56.8–89.0)	78.0 (58.3–87.8)	79.0 (60.0–87.0)
GOLD I (Mild)	13/34 (38.2)	14/28 (50.0)	13/27 (48.2)
GOLD II (Moderate)	18/34 (53.0)	11/28 (39.3)	13/27 (48.2)
GOLD III (Severe)	2/34 (5.9)	3/28 (10.7)	1/27 (3.7)
GOLD IV (Very severe)	1/34 (2.9)	0/28 (0.0)	
Surgical data			
AVR	12/34 (35.3)	7/28 (25.0)	7/27 (26.0)
CABG	20/34 (58.8)	14/28 (50.0)	15/27 (55.6)
CABG + AVR	2/34 (5.9)	7/28 (25.0)	5/27 (18.5)
Surgical time, min.	197.4 ± 67.2	197.6 ± 75.8	200.3 ± 36.3
Cardiopulmonary bypass time, min.	98.1 ± 42.7	102.7 ± 54.7	101.2 ± 27.4
Aortic cross-clamp time, min.	61.7 ± 33.4	59.5 ± 38.3	63.8 ± 18.9
Gain volume, L	1.2 ± 0.9	1.6 ± 0.9	1.2 ± 1.0

Abbreviations: CPB, cardiopulmonary bypass; SD, standard deviation; AMI, acute myocardial infarction; ASA, American Society of Anesthesiologists; AVR, aortic valve replacement; CABG, coronary artery bypass graft; COPD, chronic obstructive pulmonary disease; FEV1, forced expiratory volume in 1 second; GOLD, global initiative for chronic obstructive lung disease; HTK, histidine-tryptophan-ketoglutarate; ICU, intensive care unit; IQR, interquartile range. ^†^ Determined by preoperative echocardiography. ^‡^ Chronic heart failure, New York heart association Class III or worse. ^§^ Recent AMI <3 months prior to the surgery. Insulin and non-insulin dependent diabetes mellitus. ** Estimated creatinine clearance by Cockcroft–Gault equation in mL/min/1.73 m^2^. ^††^ Excess alcohol consumption, >14 for women and >21 for men units’ alcohol per week.

**Table 2 jcm-07-00462-t002:** Results of arterial blood gas analyses and interleukin-6 in samples collected before CPB, at the end of CPB, and 2, 4, 6, and 20 h post-CPB.

		Pre-CPB			End CPB			2 Hours Post-CPB			4 Hours Post-CPB			6 Hours Post-CPB			20 Hours Post-CPB			**Time (*p*-value)**	**Time -Group (*p*-value)**
		Standard	HTK	O_2_	Standard	HTK	O_2_	Standard	HTK	O_2_	Standard	HTK	O_2_	Standard	HTK	O_2_	Standard	HTK	**O_2_**		
pH	Mean	7.39	7.40	7.40	7.39	7.34	7.37	7.32	7.29	7.32	7.31	7.28	7.32	7.32	7.29	7.32	7.37	7.38	7.37	0.15	0.38
	SD	0.04	0.05	0.04	0.04	0.05	0.04	0.04	0.06	0.08	0.06	0.05	0.08	0.05	0.04	0.08	0.05	0.04	0.06		
pCO_2_ (kPa)	Mean	5.6	5.6	5.4	4.8	5.1	5.0	5.9	5.9	5.7	6.0	6.0	5.6	5.8	5.7	5.6	5.4	5.2	5.2	3.81 × 10^16^	0.87
	SD	0.6	0.8	0.6	0.5	0.7	0.5	0.7	1.0	1.2	0.7	0.7	1.1	0.6	0.7	0.9	0.8	0.6	0.5		
pO_2_ (kPa)	Mean	44.8	35.5	40.6	34.4	22.5	31.7	18.0	19.9	19.7	14.9	15.0	18.7	14.2	13.7	15.8	12.4	11.5	13.5	7.82 × 10^89^	0.04
	SD	12.6	9.2	14.5	14.4	11.6	14.2	6.5	7.3	6.9	3.8	4.3	8.6	3.3	3.0	3.2	2.6	2.0	2.8		
sO_2_ (%)	Mean	99.7	99.8	99.6	99.6	98.4	99.0	98.2	98.3	98.2	96.7	97.6	98.3	96.7	97.1	97.9	97.3	96.9	97.9	0.01	0.27
	SD	0.7	0.3	1.1	1.0	2.1	1.9	2.0	1.6	2.3	6.8	1.6	2.7	5.8	2.3	2.6	2.1	1.7	1.8		
Hb (mmol/L)	Mean	7.6	7.6	7.3	5.8	5.9	5.6	6.9	7.2	6.6	7.0	7.2	6.6	7.0	7.1	6.5	6.6	6.6	5.6	6.49 × 10^31^	0.04
	SD	0.9	0.9	0.9	0.8	0.8	0.6	0.9	0.9	0.6	1.0	0.9	0.7	0.9	1.0	1.0	1.0	0.8	1.3		
Hct (%)	Mean	37.6	37.6	36.2	28.9	29.3	28.1	34.4	35.7	33.0	34.8	35.9	32.7	34.8	35.0	32.3	33.0	33.0	29.0	9.45 × 10^30^	0.17
	SD	4.6	4.6	4.4	3.8	3.8	3.0	4.1	4.2	3.0	4.7	4.1	3.5	4.5	5.0	4.7	4.7	4.1	4.0		
K^+^ (mmol/L)	Mean	3.9	3.8	3.8	5.1	5.3	4.8	5.1	5.1	4.7	4.8	4.7	4.5	4.7	4.6	4.4	4.4	4.2	4.4	7.89 × 10^39^	0.006
	SD	0.2	0.3	0.4	0.6	0.5	0.4	0.5	0.6	0.5	0.5	0.5	0.6	0.7	0.6	0.5	0.4	0.4	0.3		
Na^+^ (mmol/L)	Mean	137.5	138.1	137.8	134.8	129.9	135.2	135.9	133.0	136.8	136.7	135.2	137.7	137.4	136.5	138.8	137.5	137.4	138.8	0.002	0.01
	SD	2.6	2.2	3.2	2.4	3.0	2.0	2.8	2.4	2.4	3.0	3.1	2.8	2.7	3.2	2.6	2.9	2.0	4.0		
Ca^++^ (mmol/L)	Mean	1.18	1.17	1.17	1.22	1.24	1.25	1.23	1.21	1.23	1.18	1.19	1.19	1.17	1.17	1.17	1.13	1.13	1.13	5.97 × 10^10^	0.98
	SD	0.06	0.04	0.04	0.18	0.11	0.12	0.09	0.08	0.08	0.07	0.07	0.07	0.06	0.05	0.06	0.05	0.05	0.05		
HCO_3_^−^ (mEq/L)	Mean	24.5	25.1	24.7	22.1	20.3	21.8	21.8	19.9	21.1	21.2	19.8	20.9	21.4	19.8	20.8	22.8	22.9	22.8	1.93 × 10^32^	1.96 × 10^8^
	SD	1.6	1.6	1.7	1.8	1.4	1.6	1.9	1.5	1.7	2.4	1.4	1.9	2.0	1.7	2.2	2.0	2.0	2.2		
SBE	Mean	0.2	0.8	0.3	−3.1	−5.0	−3.2	−2.9	−5.1	−3.8	−3.5	−5.1	−4.0	−3.3	−5.4	−4.2	−1.8	−1.9	−2.0	3.14 × 10^44^	0.0003
	SD	1.9	1.8	2.0	2.2	1.7	1.9	2.3	1.9	1.8	3.0	1.8	2.1	2.4	2.2	2.7	2.5	2.5	2.7		
Glucose (mmol/L)	Mean	6.4	6.3	6.4	6.9	7.4	7.5	7.3	8.3	8.2	8.6	9.2	9.2	9.0	9.2	8.7	7.7	8.0	8.0	3.36 × 10^23^	0.84
	SD	1.6	1.4	1.5	1.8	1.5	1.6	1.7	2.6	1.6	2.0	2.5	2.0	1.6	1.5	1.5	2.1	1.4	1.6		
Lactate (mmol/L)	Mean	0.7	0.7	0.8	1.6	1.8	1.9	1.2	1.4	1.4	1.5	1.9	1.9	1.7	2.0	2.2	1.3	1.3	2.0	1.49 × 10^9^	0.70
	SD	0.3	0.2	0.3	0.5	0.5	0.6	0.6	0.6	0.6	1.0	1.6	1.2	1.1	1.2	1.6	1.1	0.6	3.3		
IL-6 (pg/mL)	Mean	4.9	2.9	13.1	26.2	23.3	55.2	61.0	63.0	98.3	61.0	59.8	96.1	71.5	61.8	96.1	67.8	53.1	100.4	3.74 × 10^7^	0.45
	SD	11.6	2.3	52.4	21.9	17.0	85.0	49.4	56.9	153.2	62.5	58.8	159.7	91.7	98.5	189.4	93.1	50.2	192.0		

Two-way ANOVA (analysis of variance) and its corresponding Tukey’s post-hoc test for group comparison was used to detect differences between patients as a consequence of surgery (time) and treatment received (time-group interaction). A two-tailed *p*-value ≤0.05 was considered significant. Ca^++^, calcium; IL-6, interleukin-6; K^+^, potassium; Na^+^, sodium; pCO_2_, partial pressure of carbon dioxide; pH, acidity/alkalinity; pO_2_, partial pressure of oxygen; HCO_3_^−^, bicarbonate; SBE, base excess; sO_2_, saturation of oxygen; Hb, haemoglobin; Hct, haematocrit.

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
