# Peer review of "Lung Protection Strategies during Cardiopulmonary Bypass Affect the Composition of Blood Electrolytes and Metabolites—A Randomized Controlled Trial"

_jcm, 2018, doi:10.3390/jcm7110462_

Reviewer 1 Report

Very encompassing follow up to the Pulmonary Protection Trial. Very well written and organized with conclusions supported by the data. Congratulations on a job very well done!

I thought this was a well written sub study analysis of the randomized control trial "Pulmonary Protection Trial". The trial showed no clinical benefit to having normal bypass vs bypass with warm antegrade blood being circulated around the pulmonary vascular system vs HTK being circulated around the pulmonary vascular system with cell saver being used to scavenge the blood from the bronchial arteries. The authors were hoping to show a difference because they believe there is ischemia reperfusion injury after CPB. I, and many others do not think CPB causes IR injury since the bronchial tree is supplied by the aorta. This is only 7% of cardiac output, but is sufficient. If you look at lung transplant IR injury is fairly common because the bronchial arteries are not re-anastomosed and the lung is supplied by the pulmonary arteries only.  This paper looks at the metabolites and IL-6 between these 3 groups and makes logical conclusions. Although this is a "negative study", I find little fault with the methods, data, or conclusions made by the authors. Their "ultimate premise" was incorrect, but that had never been proven before this experiment and some may disagree still that patient do not get IR injury after prolonged CPB time.

Author Response

Thank you for your complimentary words regarding our sub-study of the Pulmonary Protection Trial.

We agree that the primary hypotheses of CPB causing an IR injury of the lungs were not supported by our findings. On the other hand, the results supports the hypotheses of standard CPB without pulmonary artery perfusion does not cause an ischemic insult of the lungs due to sufficient supply of blood via the bronchial arteries.

Reviewer 2 Report

* This sub-study of the Pulmonary Protection Trial aimed to elucidate changes in arterial blood gas analyses, inflammatory protein interleukin-6, and metabolites of 90 chronic obstructive pulmonary disease patients following two lung protective regimens of This 

This sub-study of the Pulmonary Protection Trial aimed to elucidate changes in arterial blood gas analyses, inflammatory protein interleukin-6, and metabolites of 90 chronic obstructive pulmonary disease patients following two lung protective regimens of pulmonary artery perfusion with either hypothermic Histidine-Tryptophan-Ketoglutarate (HTK) solution or normothermic oxygenated blood during CPB, compared to the standard CPB with no pulmonary perfusion

This is the first study to investigate the metabolic response in COPD patients undergoing cardiac surgery randomized to receive either pulmonary artery perfusion with oxygenated blood or HTK solution, compared to standard CPB regimen without pulmonary perfusion

Patients underwent an overnight fast, standardized anesthesia and surgical management, and were assigned to receive the standard CPB regimen or pulmonary artery perfusion, with either continuous normothermic oxygenated blood during CPB or a single-shot hypothermic of HTK solution at the same time as the first cardioplegia

Immediately after anesthetic induction, a catheter was inserted in the pulmonary artery and blood samples for (IL-6) and metabolite analysis were drawn at baseline (pre-CPB), immediately after weaning from CPB (end CPB), and the following 2, 4, 6, and 20 hours after weaning from CPB. Samples were simultaneously drawn from a catheter in arteria radialis for immediate blood gas analysis. At the end of CPB, pH levels dropped slightly in all patients, with more pronounced decreases in the HTK-receiving patients. Base excess and bicarbonate also decreased with unchanged partial pressure of carbon dioxide in accordance with the HTK driven metabolic acidosis although without changes in lactate. At 20 hours post-CPB, pH levels returned to pre-surgical values in all patients. The partial pressure of oxygen was in all patients above physiological values from pre-CPB until normalization 20 hours post-CPB. Arterial blood glucose and lactate levels increased right after CPB in all patients regardless of the treatment received, and remained elevated through the following 20 hours post-CPB. Potassium increased, sodium decreased immediately after CPB, especially in the HTK-receiving patients, and nearly returned to their baseline levels 20 hours after CPB. A steep increase of IL-6 was observed in all patients. Glycolytic metabolites, several amino acids and ketone bodies were all increased immediately after CPB in all patients. HTK-receiving patients had higher mannitol levels at the end of CPB. The protein derived N-acetyl-glycoprotein fragment, lipoproteins and fatty acids were decreased in all patients at the end of CPB. Plasma glutamate, glutamine, aspartate, threonine, methionine, creatine, glycine, alanine, format, and 2-ketoglutarate were all elevated in the HTK group compared to the standard and oxygenated blood groups. Citrate levels increased to a minor degree postoperatively in the HTK-receiving patients compared to the patients receiving oxygenated blood or standard CPB. Sodium changed more significantly in the HTK receiving patients at the end of CPB. The levels of free fatty acids, lipoproteins, monounsaturated fatty acids, triacylglycerol, and N-acetyl-glycoprotein fragments were significantly lower at the end of CPB. The TCA cycle metabolite citrate was found to be elevated immediately after CPB in all patients, with lower increases in the HTK group. Histidine and the metabolites involved in histidine catabolism were found to be significantly higher up to and including the last measurement at 20 hours post-CPB. Plasma urea increased in all patients post-CPB and the HTK receiving patients experienced elevated production 4 hours post-CPB. All patients had decreased gas exchange, augmented inflammation, and metabolite alteration during and after CPB. Patients receiving HTK solution had an excess of metabolites involved in energy production and detoxification of reactive oxygen species. Patients receiving HTK suffered a transient isotonic hyponatremia that resolved within 20 hours post-CPB. The arterial blood gas analyses, IL-6, and several metabolites changed significantly in the peri and postoperative period in cardiac surgery patients with COPD. When comparing the molecular and electrolyte differences between pulmonary artery perfusion with oxygenated blood to the standard regimen similar patterns were observed during and after CPB. Metabolite data indicate that pulmonary perfusion with HTK during CPB may provide some of the necessary substrates needed to counteract ischemia-reperfusion injury, protect against severe energy depletion, and ameliorate ROS production.

Althoughthe study is quite interesting there are several limitations that have been already underlined by the authors:

* Limited follow-up of the study: 20 hours post-CPB

* The use of pulmonary artery perfusion regimens could have prevented the development of these postoperative complications

* Clinical and metabolite outcomes were not measured at the time of maximal damage, which impairs the generalizability of the observations

* In a clinical trials with a cohort of complex critically ill patients the outcomes will be determined primarily by the severity of their underlying illness

* Cardiac surgical patients are a relatively homogeneous group, the inevitable heterogeneity may leave this sub-study underpowered to detect group differences in relation to clinical outcomes and metabolite estimates, and specifically, for the Comparison of pulmonary artery perfusion with oxygenated blood to standard CPB

* Additional studies are needed to confirm the metabolite results and further elucidate how to diminish ischemia-reperfusion injury of the lungs during CPB to reduce the risk of developing acute respiratory distress syndrome

The authors could speculate a little more on the limitation indcated above in the Discussion section.

Author Response

We thank the reviewer for outlining the limitations of our study. We have incorporated them in the discussion with the following sentences being added/changed:

The main aim of this sub-study was to investigate the molecular effects of the two pulmonary perfusion regimes compared to  standard CPB, on blood obtained from COPD patients included in the randomized Pulmonary Protection Trial [46]. In particularly we aimed to find changes in arterial blood gas, electrolytes, plasma inflammatory protein IL-6, and plasma metabolites associated with hypothesized ischemia-reperfusion injury and the received regimen. The results should be interpreted with respect to some important limitations. Although cardiac surgical patients are relative homogenous, outcomes are, besides the interventions, influenced by the individual patient’s comorbidities and may leave the study underpowered to detect any group differences. In addition, our follow-up period was limited to 20-hours post CPB, which is not the time of maximal pulmonary damage, which questions the generalizability of our results.”

“These observations indicate that the additional blood supply does not as hypothesized alter what is already achieved by the bronchial arteries. When comparing the differences between pulmonary artery perfusion with HTK solution to oxygenated blood or the standard regimen, we found that some of the HTK solution enters the systemic circuit and elicits a significant impact on arterial blood gas analyses and metabolites, with some of the effects still present 20 hours after its administration. Results confirm the development of an isotonic hyponatremia following administration of HTK solution, and that this should not be corrected without concomitant hypotonicity. In addition, although plasma IL-6 levels increased in all patients during and after CPB, HTK receiving patients had, in general, lower levels of this inflammatory protein; however, the difference was not statistically significant. Metabolite data indicate that pulmonary perfusion with HTK during CPB may provide some of the necessary substrates needed to counteract ischemia-reperfusion injury, protect against severe energy depletion, and ameliorate ROS production. Additional studies are needed to confirm the metabolite results and further elucidate how to diminish ischemia-reperfusion injury of the lungs during CPB to reduce the risk of developing acute respiratory distress syndrome.”